# Behavior of Circular CFST Columns Subjected to Different Lateral Impact Energy

**Xiaoyong Zhang [1], Yu Chen [2],\*, Xiaosheng Shen [1] and Yao Zhu [1]**

[1] School of Urban Construction, Yangtze University, Jingzhou 434023, China;
201771359@yangtzeu.edu.cn (X.Z.); 201772364@yangtzeu.edu.cn (X.S.); skinyao@163.com (Y.Z.)

[2] College of Civil Engineering, Fuzhou University, Fuzhou 350116, China

\* Correspondence: kinkingingin@163.com; Tel.: +86-180-3021-9629

**Abstract:** Forty-eight circular concrete-filled steel tube (CFST) columns subjected to lateral impact were tested to investigate the behavior of circular CFST columns under axial compressive load. Analyses of effects of concrete compressive strength, impact location and impact energy on residual ultimate axial capacity, ductility and initial stiffness are provided in this paper. It is found that lateral impact has negative effects on residual ultimate axial capacity of circular CFST columns from test results. Residual ultimate axial capacity decreases as impact energy increases and impact location comes close to the end of the specimen. It is also found that increasing concrete compressive strength can reduce the negative effects of impact location on residual ultimate axial capacity. Ductility and the initial stiffness of circular CFST columns decrease as impact energy increases. Ductility and the initial stiffness increase as impact location varies from middle-length to the end of specimens. When impact energy and impact location are constant, the ductility of the specimen with 30 MPa of concrete compressive strength is better than that of other specimens with different compressive strength. Besides, analyses of strain developments for 12 typical specimens to investigate failure modes under axial compressive load are provided in this paper. Strain developments have indicated that the steel at impact location becomes plastic faster than that at other locations. Based on the test results, a calculation formula is presented to predict the residual ultimate axial capacities of circular CFST columns subjected to lateral impact, and good agreement with experimental results has been achieved.

**Keywords:** lateral impact; impact energy; impact location; circular CFST; residual ultimate axial capacity

## 1. Introduction

With the development of engineering technology, the concrete-filled steel tube (CFST) has been widely used in seismic-resistant construction, high buildings, industrial buildings, bridge piers and offshore structures owing to its high strength, good ductility, light weight and so on [1–3]. CFST has the capacity of high ductility and large energy absorption [4], which leads to the perfect seismic resistance of CFST [5]. Many studies have been conducted to investigate the mechanical performance of CFST [6–11]. Bond strength of the CFST has been investigated by scholars [12,13]. The ultimate bearing capacity of the CFST under axial compressive load has been investigated by scholars [14–16]. Scholars also have investigated the behavior of tubular trusses and a new form of bolted connection with a precast reinforced concrete shear wall [17,18]. Moreover, the influence of cracks on chloride diffusivity in concrete has been studied by scholars Jianxin Peng et al. [19].

The damage caused by buildings subjected to impact or earthquake is enormous, and therefore it is of great importance to investigate the behavior of the structure after impact load. Zhang et al. [20]

investigated the seismic damage of spatial frames with steel beams connected to L-shaped CFST columns. There are numerous studies on the mechanical performance of the structure under the impact load. Wan, Chen and Zhang et al. [21–23] have investigated the behavior of hollow-section short-steel columns and square CFST columns subjected to impact load. Lateral impact load has negative effects on ultimate axial capacity of the test specimens. The ultimate axial capacity of test specimens decreases severely because of lateral impact load. In order to investigate the performances of CFST under lateral impact load, Aghdamy et al. [24] conducted a numerical study to simulate the test conditions. The most important parameters are the value of $t/D$ and the ratio of slenderness compared with other parameters, which influenced the response of a CFST column under lateral impact load in the study. The effects of transverse impact on the buckling behavior of a compressed column have been investigated by a numerical analysis, which is presented by Sastranegara et al. [25]. The critical condition to produce buckling was the transverse impact load, which could be found from this study's results. The behavior of concrete columns after transverse impact was studied under compression load through a numerical simulation by Thilakarathna et al. [26] who developed a non-linear explicit numerical model too. Yang et al. [27] aimed to optimize the crashworthiness of empty and foam-filled thin-walled square columns under oblique impact load. It was found that the optimal designs were generally different at different load angles for either empty or foam-filled columns. The result indicated that multi objective optimization design process including multiple load angles was a way to obtain more robust designs against oblique impact. Makarem et al. [28] provided a consistent methodology to assess the non-linear behavior of structural steel column under severe load. It also focused on the influence of the impactor mass, the impact velocity and the impact location on the behavior of the column. The behavior of hollow and concrete-filled thin-walled steel tubular members under transverse impact was investigated by Bambach [29] through experimental and theoretical methods. It can be summarized that the concrete-filled metal tubes provide a little energy absorption capacity when the bending hinge moment capacity increased. It was also found that stainless steel had a significant effect on energy absorption capacity. Bao et al. [30] presented numerical simulations of the dynamic responses and residual axial strength of reinforced concrete columns under short standoff blast conditions by utilizing LS-DYNA. The effects of design variables on the residual axial bearing capacity were investigated by an extensive parametric study with 12 columns, which showed that axial load ratio had a significant effect on the residual axial bearing capacity with a low transverse reinforcement ratio. Dong et al. [31] developed a theoretical model to evaluate the lateral strain, confining stress and axial stress in a CFST with a fiber reinforced polymer (FRP) jacket at various stages of load. Alam et al. [32] developed the finite element model of CFST columns strengthened by carbon fiber reinforced polymer (CFRP) and investigated the effect of a CFRP-bound length on strength under transverse impact load. The test results showed that CFRP with an effective bound length improved the strength of the CFST column and the column strengthened by CFRP could successfully avoid buckling failure. Shakir et al. [33] investigated the dynamic response of steel tube columns filled with normal or recycled aggregate concrete under lateral projectile impact. They also studied the influences of CFRP jackets on the structural behavior of those columns. The results indicated that normal and recycled aggregate concrete filled steel tube (NACFST and RACFST) specimens had a similar displacement shape. The results also indicated that the confinement of the CFRP reduced the global displacement of both the RACFST and NACFST specimens. Wang et al. [34] investigated blast resistance and the residual strength of CFST columns with the damage of close-range blast loads. The results indicated that CFST columns still had a large portion of axial load capacity even after close-range blast events.

Given the research mentioned above, the mechanical behavior of columns under lateral impact load has been investigated with many numerical and test researches [35,36]. A conclusion can be draw that CFST columns have great capacity of the lateral impact resistance. Nevertheless, there is little theoretical research and few calculation formulae about the residual ultimate axial capacity of the circular CFST columns after the lateral impact damage. This paper mainly investigates the effects

of concrete compressive strength, impact location and impact energy on the residual ultimate axial capacity of the circular CFST columns. According to the experimental results, a calculation formula is proposed to predict the residual ultimate axial capacity of the specimens after the lateral impact load, which has been verified to be accurate.

## 2. Experimental Investigation

### 2.1. Material Properties

#### 2.1.1. Steel Tube

Chinese Standard Q235 steel with the yield strength of 235 MPa was used to fabricate steel tubes in the study. The yield strength ($f_y$) and tensile strength ($f_u$) of steel tube used in the test, were tested according to the metallic materials (GB/T 228-2002) of Chinese Standard. The test results of three tensile coupons, which are labeled as S1, S2 and S3, are shown in Table 1.

**Table 1.** Properties of steel tube.

| Test Coupons | Yield Strength $f_y$ (MPa) | Average Yield Strength $f_y$ (MPa) | Ultimate Strength $f_u$ (MPa) | Average Ultimate Strength $f_u$ (MPa) | $f_y/f_u$ |
|---|---|---|---|---|---|
| S1 | 272 | | 353 | | |
| S2 | 263 | 264 | 342 | 352 | 0.75 |
| S3 | 257 | | 361 | | |

#### 2.1.2. Concrete

Concrete compressive strength used in the test includes 20 MPa, 30 MPa and 40 MPa. The different mixed proportions of concrete are presented in Table 2. Nine $150 \times 150 \times 150$ mm cubes, which were cast and cured under the same conditions of the concrete used in the tests, were used to investigate concrete compressive strength ($f_{cu}$). The concrete compressive strength is shown in Table 3.

**Table 2.** Mixed details of different concrete compressive strength.

| Concrete Compressive Strength | Cement (kg) | Sand (kg) | Coarse Aggregate (kg) | Water (kg) |
|---|---|---|---|---|
| 20 MPa | 350 | 690 | 1160 | 185 |
| 30 MPa | 450 | 600 | 1192 | 183 |
| 40 MPa | 520 | 525 | 1220 | 178 |

Ps: These mixtures were used for one-meter cubes concrete.

**Table 3.** Properties of concrete.

| Nominal Concrete Compressive Strength (C) | Test Coupons | Concrete Compressive Strength $f_{cu}$ (MPa) | Average Concrete Compressive Strength $f_{cu}$ (MPa) |
|---|---|---|---|
| 20 (MPa) | No.1 | 20.2 | |
| | No.2 | 23.6 | 22.13 |
| | No.3 | 22.6 | |
| 30 (MPa) | No.1 | 32.4 | |
| | No.2 | 31.6 | 32.70 |
| | No.3 | 34.1 | |
| 40 (MPa) | No.1 | 44.2 | |
| | No.2 | 43.5 | 43.13 |
| | No.3 | 41.7 | |

### 2.2. Test Specimens

The test program is designed to investigate the influence of the parameters on the residual ultimate axial capacity of circular CFST columns subjected to lateral impact. A total of 48 specimens

were fabricated in this test. Three specimens were tested for comparison purpose, and the remaining 45 specimens were tested to evaluate the effect of lateral impact on residual ultimate axial capacity. Details of specimens are shown in Table 4. The height (*H*) of all specimens is 300 mm and the external diameter (*D*) of cross section of corresponding specimen is 89 mm. Steel tube thickness (*T*) used in the test is 4 mm. Regarding to the engineering problem of impact for the CFST columns, the investigated parameters of impact location and impact energy are set to simulate the CFST columns under different impact damage. The impact resistance and failure modes of the CFST columns are investigated to provide some possible method for the reinforcement of the columns. The parameters of concrete compressive strength are also investigated to give some advice for the choice of concrete used in civil engineering project. All tested specimens are labeled according to the parameters so that the specimens can be easily identified. For example, specimen C20-L0.25-E7500 is a steel tube filled with concrete (the concrete compressive strength is 20 MPa of this specimen), which is subjected to 7500 *J* impact energy at 0.25 height. The detailed parameters in the study include:

- Concrete compressive strength (*C*): 20 MPa, 30 MPa and 40 MPa;
- Impact location (*L*): 0.50 *H*, 0.33 *H* and 0.25 *H*;
- Impact energy (*E*): 5000 *J*, 7500 *J*, 10,000 *J*, 12,500 *J* and 15,000 *J*.

**Table 4.** Details of the specimens.

| Specimen | $N^P$ (MPa) | H (mm) | D (mm) | T (mm) | H/D | D/T | C (MPa) | L (H) | E (J) |
|---|---|---|---|---|---|---|---|---|---|
| C20-L0-E0 | 490 | 300 | 89 | 4 | 3.37 | 22.25 | 20 | 0 | 0 |
| C20-L0.50-E5000 | 490 | 300 | 89 | 4 | 3.37 | 22.25 | 20 | 0.50 | 5000 |
| C20-L0.50-E7500 | 490 | 300 | 89 | 4 | 3.37 | 22.25 | 20 | 0.50 | 7500 |
| C20-L0.50-E10000 | 490 | 300 | 89 | 4 | 3.37 | 22.25 | 20 | 0.50 | 10,000 |
| C20-L0.50-E12500 | 490 | 300 | 89 | 4 | 3.37 | 22.25 | 20 | 0.50 | 12,500 |
| C20-L0.50-E15000 | 490 | 300 | 89 | 4 | 3.37 | 22.25 | 20 | 0.50 | 15,000 |
| C20-L0.33-E5000 | 490 | 300 | 89 | 4 | 3.37 | 22.25 | 20 | 0.33 | 5000 |
| C20-L0.33-E7500 | 490 | 300 | 89 | 4 | 3.37 | 22.25 | 20 | 0.33 | 7500 |
| C20-L0.33-E10000 | 490 | 300 | 89 | 4 | 3.37 | 22.25 | 20 | 0.33 | 10,000 |
| C20-L0.33-E12500 | 490 | 300 | 89 | 4 | 3.37 | 22.25 | 20 | 0.33 | 12,500 |
| C20-L0.33-E15000 | 490 | 300 | 89 | 4 | 3.37 | 22.25 | 20 | 0.33 | 15,000 |
| C20-L0.25-E5000 | 490 | 300 | 89 | 4 | 3.37 | 22.25 | 20 | 0.25 | 5000 |
| C20-L0.25-E7500 | 490 | 300 | 89 | 4 | 3.37 | 22.25 | 20 | 0.25 | 7500 |
| C20-L0.25-E10000 | 490 | 300 | 89 | 4 | 3.37 | 22.25 | 20 | 0.25 | 10,000 |
| C20-L0.25-E12500 | 490 | 300 | 89 | 4 | 3.37 | 22.25 | 20 | 0.25 | 12,500 |
| C20-L0.25-E15000 | 490 | 300 | 89 | 4 | 3.37 | 22.25 | 20 | 0.25 | 15,000 |
| C30-L0-E0 | 544 | 300 | 89 | 4 | 3.37 | 22.25 | 30 | 0 | 0 |
| C30-L0.50-E5000 | 544 | 300 | 89 | 4 | 3.37 | 22.25 | 30 | 0.50 | 5000 |
| C30-L0.50-E7500 | 544 | 300 | 89 | 4 | 3.37 | 22.25 | 30 | 0.50 | 7500 |
| C30-L0.50-E10000 | 544 | 300 | 89 | 4 | 3.37 | 22.25 | 30 | 0.50 | 10,000 |
| C30-L0.50-E12500 | 544 | 300 | 89 | 4 | 3.37 | 22.25 | 30 | 0.50 | 12,500 |
| C30-L0.50-E15000 | 544 | 300 | 89 | 4 | 3.37 | 22.25 | 30 | 0.50 | 15,000 |
| C30-L0.33-E5000 | 544 | 300 | 89 | 4 | 3.37 | 22.25 | 30 | 0.33 | 5000 |
| C30-L0.33-E7500 | 544 | 300 | 89 | 4 | 3.37 | 22.25 | 30 | 0.33 | 7500 |
| C30-L0.33-E10000 | 544 | 300 | 89 | 4 | 3.37 | 22.25 | 30 | 0.33 | 10,000 |
| C30-L0.33-E12500 | 544 | 300 | 89 | 4 | 3.37 | 22.25 | 30 | 0.33 | 12,500 |
| C30-L0.33-E15000 | 544 | 300 | 89 | 4 | 3.37 | 22.25 | 30 | 0.33 | 15,000 |
| C30-L0.25-E5000 | 544 | 300 | 89 | 4 | 3.37 | 22.25 | 30 | 0.25 | 5000 |
| C30-L0.25-E7500 | 544 | 300 | 89 | 4 | 3.37 | 22.25 | 30 | 0.25 | 7500 |
| C30-L0.25-E10000 | 544 | 300 | 89 | 4 | 3.37 | 22.25 | 30 | 0.25 | 10,000 |
| C30-L0.25-E12500 | 544 | 300 | 89 | 4 | 3.37 | 22.25 | 30 | 0.25 | 12,500 |
| C30-L0.25-E15000 | 544 | 300 | 89 | 4 | 3.37 | 22.25 | 30 | 0.25 | 15,000 |
| C40-L0-E0 | 598 | 300 | 89 | 4 | 3.37 | 22.25 | 40 | 0 | 0 |
| C40-L0.50-E5000 | 598 | 300 | 89 | 4 | 3.37 | 22.25 | 40 | 0.50 | 5000 |
| C40-L0.50-E7500 | 598 | 300 | 89 | 4 | 3.37 | 22.25 | 40 | 0.50 | 7500 |
| C40-L0.50-E10000 | 598 | 300 | 89 | 4 | 3.37 | 22.25 | 40 | 0.50 | 10,000 |
| C40-L0.50-E12500 | 598 | 300 | 89 | 4 | 3.37 | 22.25 | 40 | 0.50 | 12,500 |
| C40-L0.50-E15000 | 598 | 300 | 89 | 4 | 3.37 | 22.25 | 40 | 0.50 | 15,000 |
| C40-L0.33-E5000 | 598 | 300 | 89 | 4 | 3.37 | 22.25 | 40 | 0.33 | 5000 |

**Table 4.** *Cont.*

| Specimen | $N^P$ (MPa) | $H$ (mm) | $D$ (mm) | $T$ (mm) | $H/D$ | $D/T$ | $C$ (MPa) | $L$ ($H$) | $E$ ($J$) |
|---|---|---|---|---|---|---|---|---|---|
| C40-L0.33-E7500 | 598 | 300 | 89 | 4 | 3.37 | 22.25 | 40 | 0.33 | 7500 |
| C40-L0.33-E10000 | 598 | 300 | 89 | 4 | 3.37 | 22.25 | 40 | 0.33 | 10,000 |
| C40-L0.33-E12500 | 598 | 300 | 89 | 4 | 3.37 | 22.25 | 40 | 0.33 | 12,500 |
| C40-L0.33-E15000 | 598 | 300 | 89 | 4 | 3.37 | 22.25 | 40 | 0.33 | 15,000 |
| C40-L0.25-E5000 | 598 | 300 | 89 | 4 | 3.37 | 22.25 | 40 | 0.25 | 5000 |
| C40-L0.25-E7500 | 598 | 300 | 89 | 4 | 3.37 | 22.25 | 40 | 0.25 | 7500 |
| C40-L0.25-E10000 | 598 | 300 | 89 | 4 | 3.37 | 22.25 | 40 | 0.25 | 10,000 |
| C40-L0.25-E12500 | 598 | 300 | 89 | 4 | 3.37 | 22.25 | 40 | 0.25 | 12,500 |
| C40-L0.25-E15000 | 598 | 300 | 89 | 4 | 3.37 | 22.25 | 40 | 0.25 | 15,000 |

*2.3. Impact Test*

In impact test, an air hammer was employed to apply the impact load as planned, which is shown in Figure 1. Impact energy 2500 *J* for the hammer with a weight of 150 kg was applied to the specimen one time. One platform was designed to avoid this situation of the specimen subjected to impact load on two opposed surfaces during the impact test. In other words, this platform, used to guarantee that only one surface of specimen was subjected to impact load, was installed below the air hammer. When the impact load was applied to the specimen, the specimen was fixed on the platform, as shown in Figure 2.

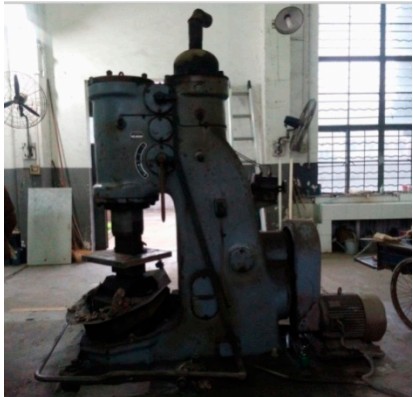

**Figure 1.** Air hammer.

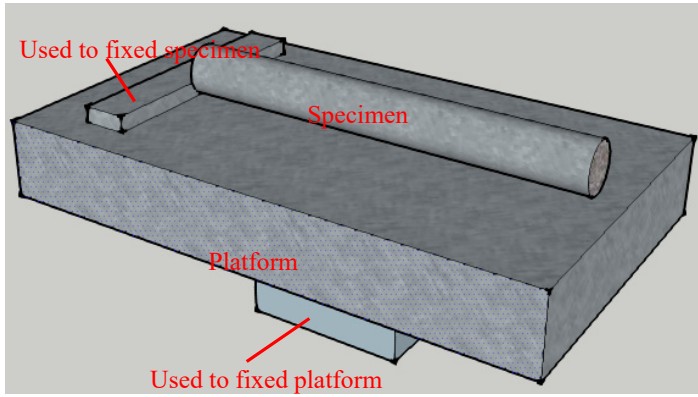

**Figure 2.** Platform used to fix the specimen.

After the impact test, different degrees of damage were found on specimens because of different concrete compressive strength, impact energy and impact location. Specifically, the deformation and damage of the specimen subjected to lateral impact became more serious when impact energy increased, impact location came close to the end of the specimen, or concrete compressive strength decreased. Deformation of specimens subjected to the lateral impact load is shown in Figure 3.

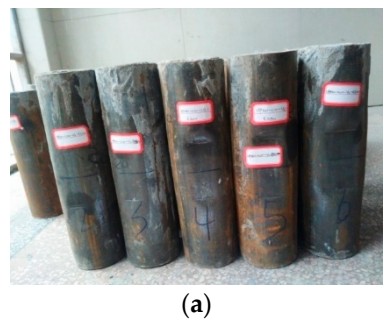 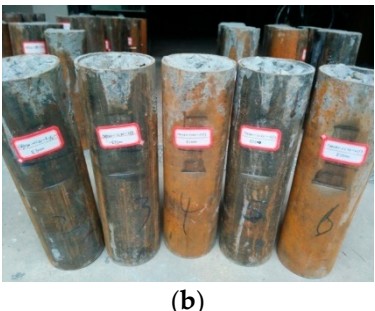 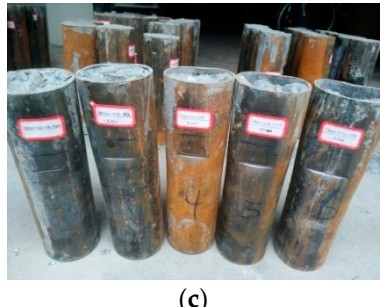

(**a**)      (**b**)      (**c**)

**Figure 3.** Specimens after lateral impact: (**a**) 0.50 *H* impact location; (**b**) 0.33 *H* impact location; (**c**) 0.25 *H* impact location.

### 2.4. Axial Compression Test

In order to observe the behavior of circular CFST columns with the damage of lateral impact under axial load adequately, the ultimate axial capacity of the specimens has been estimated before the axial compression test. The estimated ultimate axial capacity is calculated as the following Equation (1) probably:

$$N^{\mathrm{P}} = f_{\mathrm{y}} \times A_{\mathrm{s}} + f_{\mathrm{cu}} \times A_{\mathrm{c}} \tag{1}$$

where $A_{\mathrm{s}}$ is area of steel and $A_{\mathrm{c}}$ refers to area of concrete in the cross section of circular CFST columns. The estimated ultimate axial capacity of circular CFST columns is 490 kN, 544 kN and 598 kN when the concrete compressive strength is 20 MPa, 30 MPa and 40 MPa, respectively.

An electro hydraulic servo universal testing setup (the ultimate capacity of the setup is 1000 kN) was used to carry out the axial compression test. Before the axial compression tests, 12 specimens (C20-L0-E0, C30-L0-E0, C40-L0-E0, C20-L0.50-E10000, C30-L0.50-E5000, C30-L0.50-E7500, C30-L0.50-E10000, C30-L0.50-E12500, C30-L0.50-E15000, C30-L0.25-E10000, C30-L0.33-E10000, C40-L0.50-E10000) were selected to collect strain data to investigate strain development. Before strain gauges were pasted to specimens, the rust or paint of the outer surface of these 12 specimens had been cleaned up. Three measurement points were selected to paste transverse strain gauges and another three measurement points were selected for the vertical strain gauges for each specimen. For each specimen, the impact location, the side of the impact location and the back of the impact location were surface-bonded with two strain gauges. A total of six strain gauges was connected to a DH3816 automatic data acquisition system, which was used to collect strain data. This test setup is shown in Figure 4 and the arrangement of the strain gauges is shown in Figure 5.

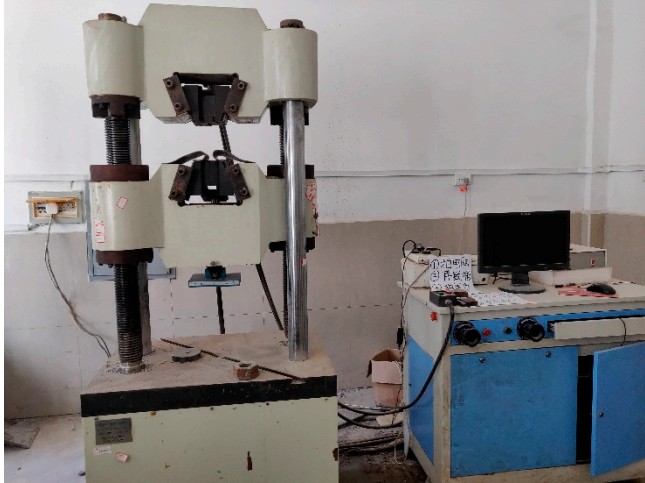

**Figure 4.** Electro hydraulic servo universal testing setup.

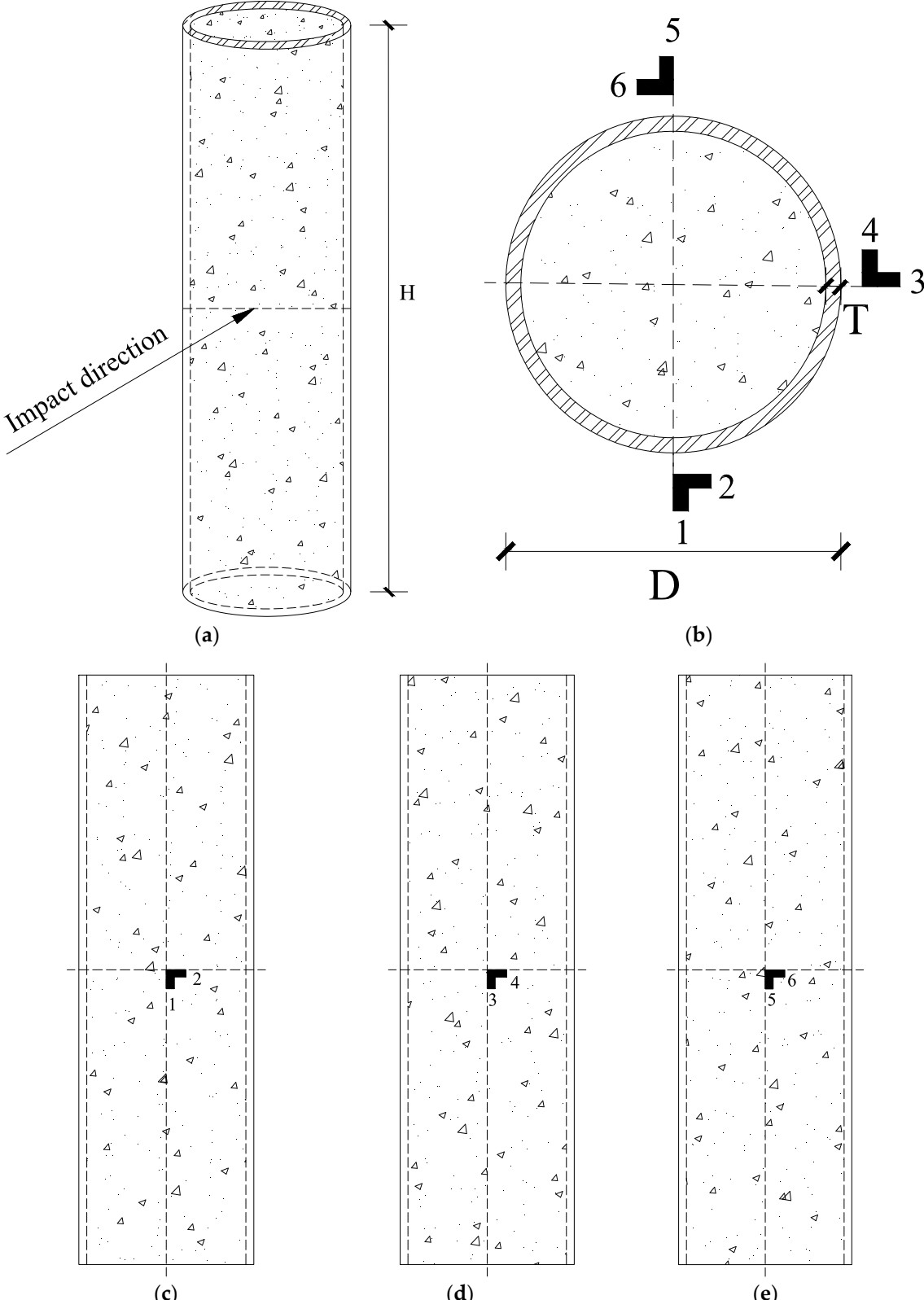

**Figure 5.** Strain gauges arrangement: (**a**) full view of the specimen; (**b**) top view of the strain gauges' arrangement; (**c**) impact point; (**d**) side point; (**e**) opposite point.

Based on estimated ultimate axial capacity of circular CFST columns, the axial compressive load was applied to the specimen under a multi-stage load. Specifically, axial compressive load was

increased gradually by 1/10 of the estimated value in each stage load when the material was within the elastic stage. Then, compressive load was gradually increased by 1/20 of the estimated value in each stage load when the material was within the elastic-plastic stage. At the end of the compression test, two methods were applied to confirm the failure of the specimen: (1) when displacement reached 10% of the height of the specimen, the specimen failed; (2) when bearing capacity increased to the maximum value and then decreased, the specimen failed.

## 3. Test Results

Influence coefficient of initial stiffness $k_c$ [37] is introduced to simplify the analysis process in this paper, which is calculated with Equation (2):

$$k_c = \frac{(E_{SC} \times A_{SC})_{impact}}{E_{SC} \times A_{SC}} \tag{2}$$

where, $E_{sc}$ is composite elastic modulus of specimens; $A_{sc}$ is cross sectional area of specimens; $(E_{sc} \times A_{sc})_{impact}$ is composite axial compression stiffness of specimen with the damage of lateral impact and $E_{sc} \times A_{sc}$ is composite axial compression stiffness of the undamaged specimens.

Ductility and initial stiffness are obtained by this method, which is called the secant stiffness method. Specifically, the ductility coefficient ($\mu$) is calculated as follows [38]:

$$\mu = \Delta_u / \Delta_u$$

where $\Delta_u$ represents ultimate deformation of specimen under axial compressive load and $\Delta_y$ represents yield deformation of corresponding specimen. To investigate the effects of lateral impact and different concrete strength on residual ultimate axial capacity, ratio $\rho$ is introduced to represent the ratio of residual ultimate axial capacity of the specimen with impact damage ($N_r$) to that of the specimen without impact damage ($N_r'$). The ratio $\rho$ can be calculated as follows: $\rho = N_r / N_r'$. Results on the residual ultimate axial capacity, ratio $\rho$, ultimate deformation, yield deformation, ductility coefficient, influence coefficient of initial stiffness, and failure mode of specimen are shown in Table 5.

**Table 5.** Test results of all specimens.

| Specimen | $N_r$ (kN) | $N^P$ (MPa) | $f_y$ (MPa) | $f_u$ (MPa) | $f_{cu}$ (MPa) | $\rho$ | $k_c$ | $\Delta_u$ (mm) | $\Delta_y$ (mm) | $\mu$ | Failure Modes |
|---|---|---|---|---|---|---|---|---|---|---|---|
| C20-L0-E0 | 594.10 | 490 | 264 | 352 | 22.13 | 1.00 | 1.00 | 30.00 | 8.49 | 3.53 | - |
| C20-L0.50-E5000 | 500.15 | 490 | 264 | 352 | 22.13 | 0.84 | 0.94 | 16.70 | 7.78 | 2.15 | mode 3 |
| C20-L0.50-E7500 | 510.00 | 490 | 264 | 352 | 22.13 | 0.86 | 0.86 | 16.15 | 6.76 | 2.39 | mode 3 |
| C20-L0.50-E10000 | 500.05 | 490 | 264 | 352 | 22.13 | 0.84 | 0.73 | 19.11 | 7.36 | 2.60 | mode 2 |
| C20-L0.50-E12500 | 550.20 | 490 | 264 | 352 | 22.13 | 0.93 | 0.92 | 16.72 | 5.84 | 2.86 | mode 3 |
| C20-L0.50-E15000 | 490.20 | 490 | 264 | 352 | 22.13 | 0.83 | 0.78 | 15.54 | 6.69 | 2.32 | mode 3 |
| C20-L0.33-E5000 | 523.75 | 490 | 264 | 352 | 22.13 | 0.88 | 0.65 | 17.23 | 7.77 | 2.22 | mode 4 |
| C20-L0.33-E7500 | 500.30 | 490 | 264 | 352 | 22.13 | 0.84 | 0.58 | 19.73 | 7.37 | 2.68 | mode 1 |
| C20-L0.33-E10000 | 550.10 | 490 | 264 | 352 | 22.13 | 0.93 | 1.36 | 19.47 | 5.52 | 3.53 | mode 3 |
| C20-L0.33-E12500 | 475.25 | 490 | 264 | 352 | 22.13 | 0.80 | 0.52 | 19.74 | 7.85 | 2.51 | mode 2 |
| C20-L0.33-E15000 | 500.25 | 490 | 264 | 352 | 22.13 | 0.84 | 0.64 | 13.35 | 6.46 | 2.07 | mode 2 |
| C20-L0.25-E5000 | 500.30 | 490 | 264 | 352 | 22.13 | 0.84 | 0.68 | 15.57 | 7.60 | 2.05 | mode 2 |
| C20-L0.25-E7500 | 460.25 | 490 | 264 | 352 | 22.13 | 0.77 | 0.67 | 17.62 | 7.98 | 2.21 | mode 4 |
| C20-L0.25-E10000 | 468.70 | 490 | 264 | 352 | 22.13 | 0.79 | 0.65 | 18.15 | 6.53 | 2.78 | mode 1 |
| C20-L0.25-E12500 | 385.20 | 490 | 264 | 352 | 22.13 | 0.65 | 0.47 | 17.37 | 7.77 | 2.34 | mode 1 |
| C20-L0.25-E15000 | 337.85 | 490 | 264 | 352 | 22.13 | 0.57 | 0.90 | 7.47 | 3.73 | 2.00 | mode 1 |
| C30-L0-E0 | 580.00 | 544 | 264 | 352 | 32.70 | 1.00 | 1.00 | 29.73 | 6.37 | 4.67 | - |
| C30-L0.50-E5000 | 563.10 | 544 | 264 | 352 | 32.70 | 0.97 | 1.19 | 20.51 | 5.92 | 3.46 | mode 4 |
| C30-L0.50-E7500 | 640.00 | 544 | 264 | 352 | 32.70 | 1.10 | 1.26 | 22.78 | 6.68 | 3.41 | mode 2 |

**Table 5.** *Cont.*

| Specimen | $N_r$ (kN) | $N^P$ (MPa) | $f_y$ (MPa) | $f_u$ (MPa) | $f_{cu}$ (MPa) | $\rho$ | $k_c$ | $\Delta_u$ (mm) | $\Delta_y$ (mm) | $\mu$ | Failure Modes |
|---|---|---|---|---|---|---|---|---|---|---|---|
| C30-L0.50-E10000 | 540.10 | 544 | 264 | 352 | 32.70 | 0.93 | 1.13 | 29.99 | 5.94 | 5.05 | mode 3 |
| C30-L0.50-E12500 | 486.80 | 544 | 264 | 352 | 32.70 | 0.84 | 0.53 | 14.30 | 8.53 | 1.68 | mode 4 |
| C30-L0.50-E15000 | 550.00 | 544 | 264 | 352 | 32.70 | 0.95 | 0.79 | 29.15 | 8.10 | 3.60 | mode 3 |
| C30-L0.33-E5000 | 575.10 | 544 | 264 | 352 | 32.70 | 0.99 | 0.74 | 16.51 | 8.72 | 1.89 | mode 3 |
| C30-L0.33-E7500 | 370.05 | 544 | 264 | 352 | 32.70 | 0.64 | 0.37 | 14.96 | 10.30 | 1.45 | mode 4 |
| C30-L0.33-E10000 | 559.50 | 544 | 264 | 352 | 32.70 | 0.96 | 0.70 | 30.00 | 11.69 | 2.57 | mode 1 |
| C30-L0.33-E12500 | 460.00 | 544 | 264 | 352 | 32.70 | 0.79 | 0.44 | 20.52 | 9.64 | 2.13 | mode 1 |
| C30-L0.33-E15000 | 379.45 | 544 | 264 | 352 | 32.70 | 0.65 | 0.97 | 8.28 | 4.61 | 1.80 | mode 2 |
| C30-L0.25-E5000 | 566.75 | 544 | 264 | 352 | 32.70 | 0.98 | 1.21 | 15.63 | 5.72 | 2.73 | mode 1 |
| C30-L0.25-E7500 | 550.00 | 544 | 264 | 352 | 32.70 | 0.95 | 0.60 | 20.59 | 8.71 | 2.36 | mode 4 |
| C30-L0.25-E10000 | 474.95 | 544 | 264 | 352 | 32.70 | 0.82 | 0.45 | 15.66 | 8.48 | 1.85 | mode 1 |
| C30-L0.25-E12500 | 510.25 | 544 | 264 | 352 | 32.70 | 0.88 | 0.90 | 15.31 | 6.09 | 2.51 | mode 4 |
| C30-L0.25-E15000 | 479.70 | 544 | 264 | 352 | 32.70 | 0.83 | 0.59 | 16.87 | 7.17 | 2.35 | mode 4 |
| C40-L0-E0 | 595.00 | 598 | 264 | 352 | 43.13 | 1.00 | 1.00 | 27.62 | 8.58 | 3.22 | - |
| C40-L0.50-E5000 | 648.80 | 598 | 264 | 352 | 43.13 | 1.09 | 0.81 | 30.00 | 9.91 | 3.03 | mode 3 |
| C40-L0.50-E7500 | 598.95 | 598 | 264 | 352 | 43.13 | 1.01 | 2.09 | 19.44 | 5.34 | 3.64 | mode 2 |
| C40-L0.50-E10000 | 579.50 | 598 | 264 | 352 | 43.13 | 0.97 | 1.55 | 18.97 | 7.44 | 2.55 | mode 4 |
| C40-L0.50-E12500 | 576.75 | 598 | 264 | 352 | 43.13 | 0.97 | 1.22 | 18.44 | 7.94 | 2.32 | mode 2 |
| C40-L0.50-E15000 | 569.00 | 598 | 264 | 352 | 43.13 | 0.96 | 1.76 | 17.07 | 6.65 | 2.57 | mode 3 |
| C40-L0.33-E5000 | 575.25 | 598 | 264 | 352 | 43.13 | 0.97 | 1.34 | 17.29 | 6.84 | 2.53 | mode 2 |
| C40-L0.33-E7500 | 526.25 | 598 | 264 | 352 | 43.13 | 0.88 | 0.93 | 22.07 | 7.54 | 2.93 | mode 1 |
| C40-L0.33-E10000 | 600.85 | 598 | 264 | 352 | 43.13 | 1.01 | 0.96 | 24.96 | 8.09 | 3.09 | mode 2 |
| C40-L0.33-E12500 | 560.00 | 598 | 264 | 352 | 43.13 | 0.94 | 1.05 | 18.11 | 8.30 | 2.18 | mode 2 |
| C40-L0.33-E15000 | 540.10 | 598 | 264 | 352 | 43.13 | 0.91 | 0.79 | 18.48 | 9.34 | 1.98 | mode 2 |
| C40-L0.25-E5000 | 475.75 | 598 | 264 | 352 | 43.13 | 0.80 | 0.55 | 22.54 | 10.47 | 2.15 | mode 1 |
| C40-L0.25-E7500 | 650.00 | 598 | 264 | 352 | 43.13 | 1.09 | 0.75 | 25.10 | 11.94 | 2.10 | mode 3 |
| C40-L0.25-E10000 | 460.25 | 598 | 264 | 352 | 43.13 | 0.77 | 0.74 | 19.48 | 8.76 | 2.22 | mode 1 |
| C40-L0.25-E12500 | 622.15 | 598 | 264 | 352 | 43.13 | 1.05 | 0.68 | 22.26 | 12.83 | 1.73 | mode 2 |
| C40-L0.25-E15000 | 600.00 | 598 | 264 | 352 | 43.13 | 1.01 | 0.88 | 21.14 | 9.70 | 2.18 | mode 1 |

## 3.1. Failure Modes

Based on varied bulged positions, typical failure processes of circular CFST columns under axial compression tests are summarized. Four kinds of failure modes are classified in the paper, which are shown in Table 5. The relevant developing processes of the four typical failure modes are shown in Figures 6–9.

A.  **The first bulged position emerged at the end of the specimen opposed to the impact point.** Then second bulged position occurred at the impact point. Lastly, the first bulge developed a ringed bulge and the specimen failed. The failure mode is called Mode One, and the failure process is shown in Figure 6. The feature of this failure mode is that impact location is close to the end of the specimen including nine specimens with impact location at 0.25 *H* and three specimens with impact location at 0.33 *H*. For failure Mode One, residual ultimate axial capacity of these specimens is lower than that of other specimens. Loss of initial stiffness of the twelve specimens is greater than that of no damaged specimens. With the increase of axial compressive load, deformation of circular CFST columns is enlarged, leading to the failure of the column.

B.  **The first bulged position occurred at the impact point.** The second bulged position emerged at the end surface of the specimen (the surface was opposed to the impact point). The failure mode is called Mode Two, and developing the process of this failure mode is shown in Figure 7. Thirteen circular CFST columns failed in Mode Two. Deformation of the specimens after the impact test was large. When axial compressive load was applied to the specimen, the deformation was enlarged at the impact point. As axial compressive load increased, local pressure increased at the end of specimen end, so the deformation of the specimen end increased. Lastly, the deformation of the specimen end and the deformation of impact point attined the max value at the same time, and then specimen failed.

C. **The bulged position appeared at the impact point.** Then deformation of impact point was enlarged with the increase of axial compressive load. Lastly, the specimen failed because of enlarged deformation. The failure mode is called Mode Three, and the failure process is shown in Figure 8. In terms of Mode Three, 11 specimens failed, including eight specimens with impact location at 0.50 *H*, two specimens at 0.33 *H* and one specimen at 0.25 *H*. The concrete of these specimens was damaged during the impact test, and the specimen became the eccentric bending member because of the damage of impact. The loss of initial stiffness of the specimens after lateral impact was greater than that of no damaged specimens. When axial compressive load increased, deformation of impact surface was enlarged. Finally, specimen failed because of the enlarged deformation.

D. **Bulged position appeared at the specimen end opposed to impact point, and bulged position developed a ringed bulge as axial compressive load increased.** The failure mode is called Mode Four. The failure process is shown in Figure 9. Nine specimens failed in Mode Four. The most specimens of these nine specimens were subjected to lateral impact at 0.25 *H*. The damage of the end of the specimen was more serious than that of other parts of the specimen. Finally, the specimen failed because of the enlarged deformation at the specimen end.

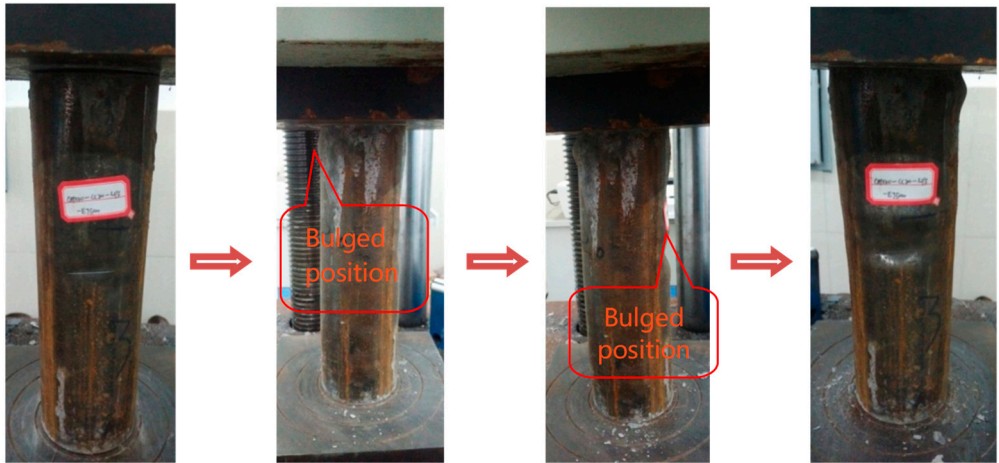

**Figure 6.** Specimen failure process as failure mode 1.

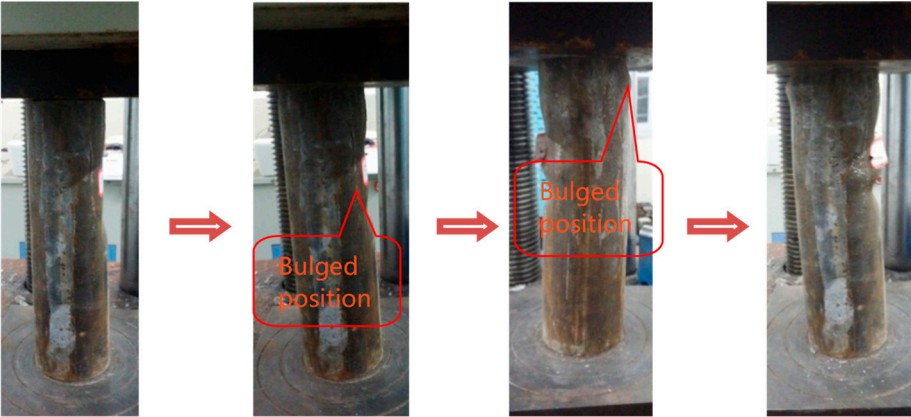

**Figure 7.** Specimen failure process as failure mode 2.

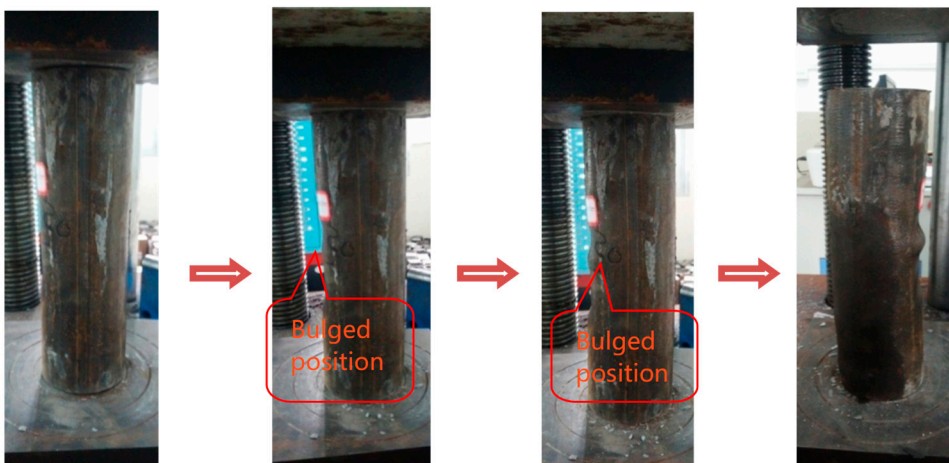

**Figure 8.** Specimen failure process as failure mode 3.

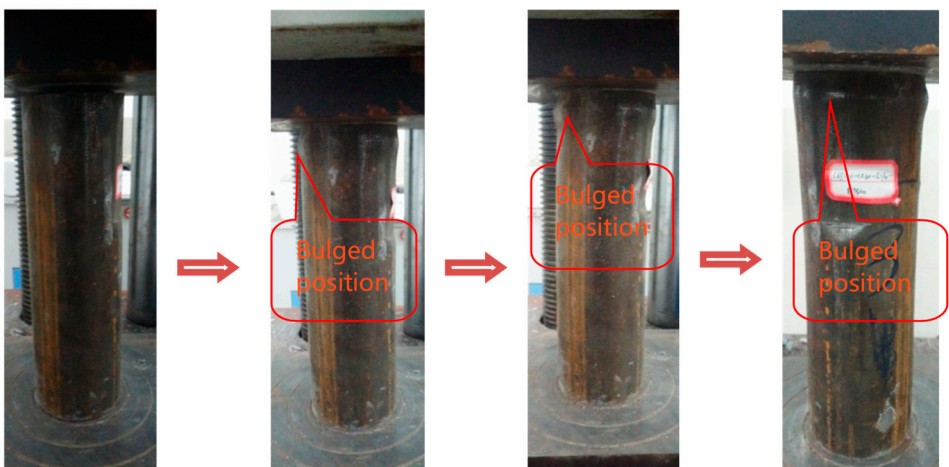

**Figure 9.** Specimen failure process as failure mode 4.

In order to meet the design requirements for impact resistance of CFST columns, the failure modes illustrate that the end of the CFST columns and the locations subjected to impact damage should be reinforced.

### 3.2. Load-Displacement Curves

Load-displacement curve data were collected by an automatic data acquisition system of the testing setup. Curves of load-displacement are presented in Figure 10. Those curves are divided into three groups ($a_1$, $a_2$, $a_3$; $b_1$, $b_2$, $b_3$ and $c_1$, $c_2$, $c_3$) according to concrete compressive strength of the specimens. The vertical axis $P$ represents the axial compressive strength and the horizontal axis $\Delta$ represents the corresponding displacement in load-displacement curve. Result show that lateral impact has negative influences on residual ultimate axial capacity, influence coefficient of initial stiffness and ductility. When the impact energy increases from 0 $J$ to 12,500 $J$ for specimens C30-L0.50, the residual ultimate axial capacity decreases from 580.00 kN to 486.80 kN. Furthermore, the specific specimen C30-L0.50-E12500 compared with those of no damaged specimen decreases by 47% in its influence coefficient of initial stiffness and a decrease by 64% in its ductility. Hence the influence coefficient of initial stiffness decreased because of lateral impact damage. The ultimate axial capacity of these specimens after lateral impact decreased seriously compared with specimens without the lateral impact damage. So there will be serious impact damages on the CFST columns in the civil engineering project. The investigation of this paper would provide the basis for CFST columns' reinforcement to improve the impact resistance.

One typical load-displacement curve is summarized from load-displacement curves and is shown in Figure 11. The load-displacement curve can be divided into three stages: elastic stage (OA), elastic-plastic stage (AB) and strengthened stage (BC). When the axial load increased within the elastic stage, the steel and concrete of the specimen bore axial load independently. With the axial load increased to elastic-plastic stage, the steel and concrete of the specimen bore axial load together. At this stage, obvious deformation can be found on the surface of the steel, which indicates that the steel entered yield stage. When axial load increases a little within the strengthened stage, the corresponding displacement increases very quickly and the specimen fails because of the excessively large displacement, which indicates that lateral impact improved the constraint effect coefficient of the CFST column.

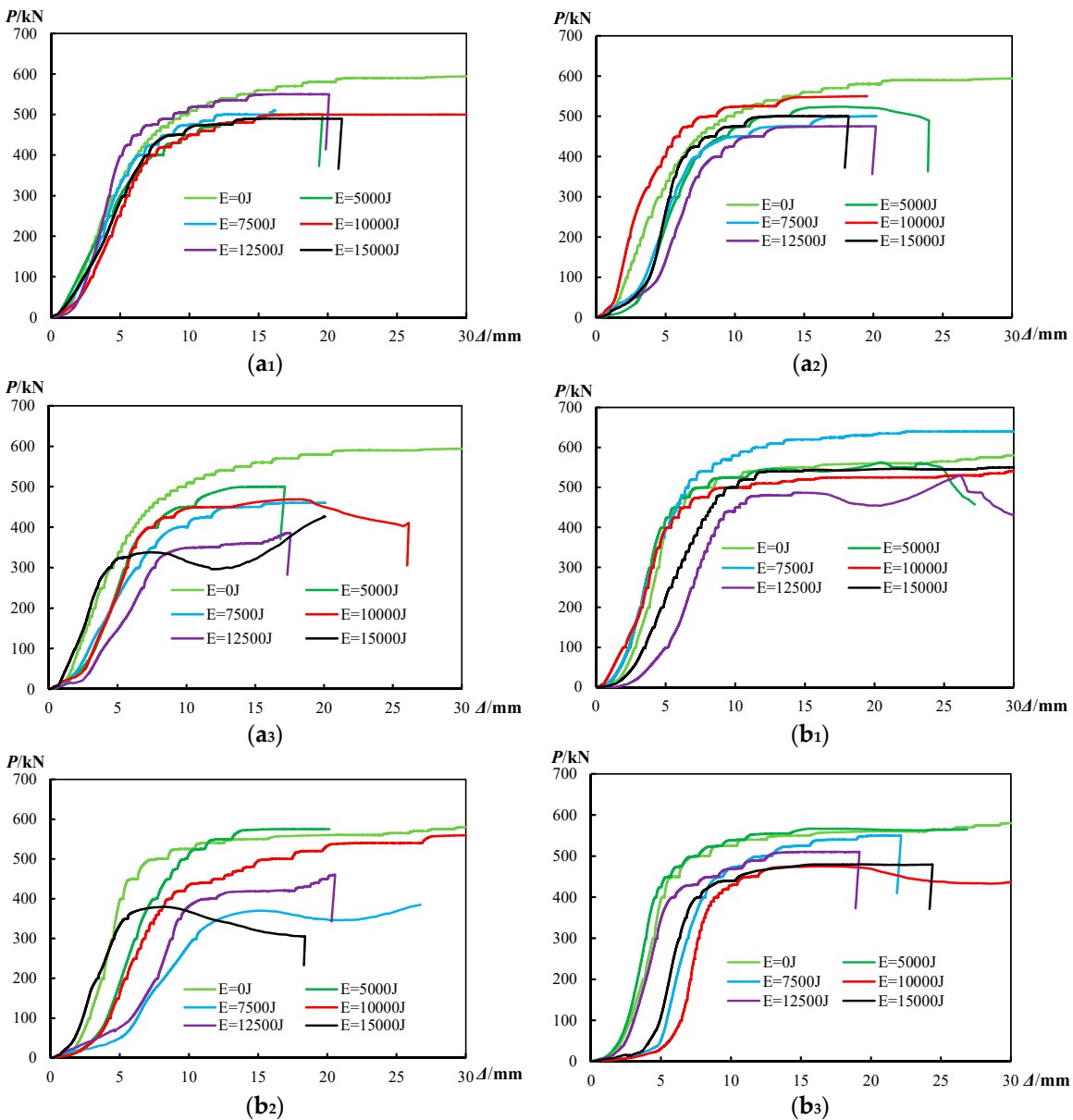

**Figure 10.** *Cont.*



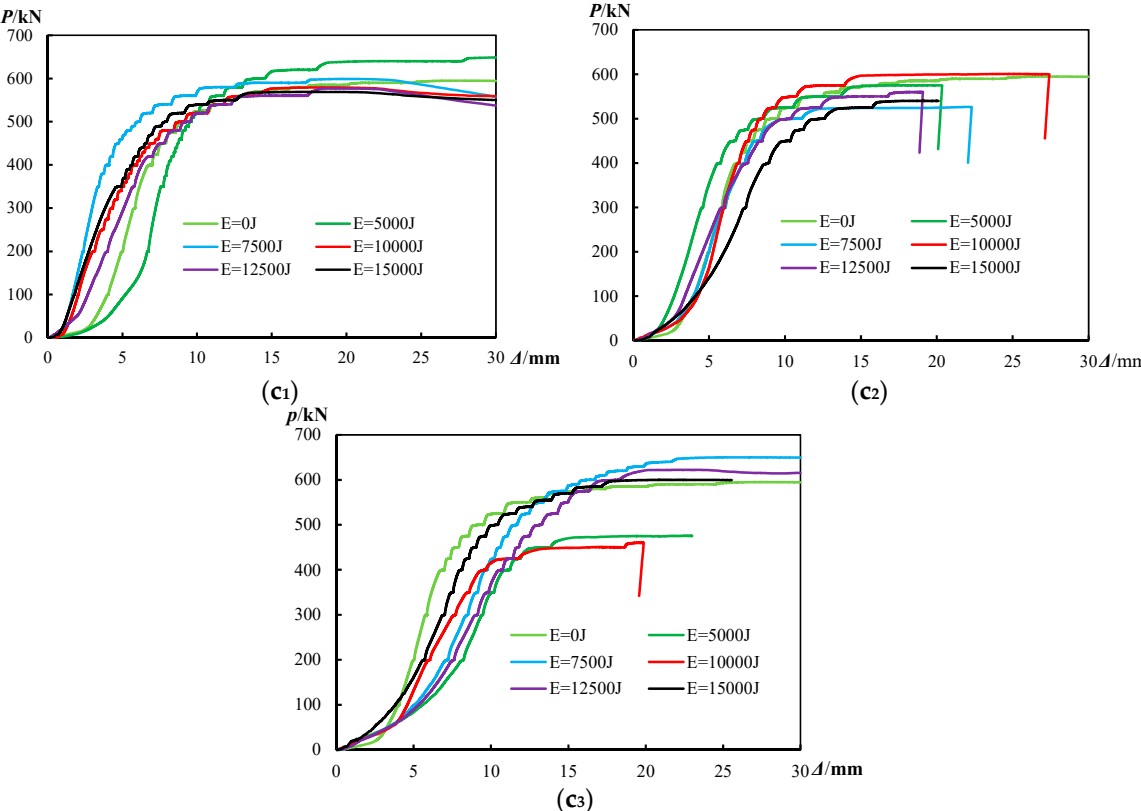

**Figure 10.** Load-displacement curves: (**a₁**) C20-L0.50; (**a₂**) C20-L0.33; (**a₃**) C20-L0.25; (**b₁**) C30-L0.50; (**b₂**) C30-L0.33; (**b₃**) C30-L0.25; (**c₁**) C40-L0.50; (**c₂**) C40-L0.33; (**c₃**) C40-L0.25.

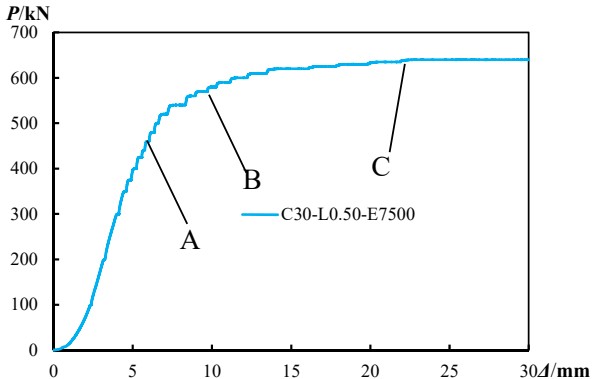

**Figure 11.** Typical load–displacement curve.

*3.3. Load–Strain Curves*

Tensile strain is defined as positive strain and compressive strain is defined as negative strain to analyze the effects of lateral impact on the residual ultimate axial capacity of the circular CFST column. In other words, the strain obtained from compressive test is positive, which indicates that there exists tensile strain. Strain obtained from the compressive test is negative, which indicates that there exists compressive strain. The vertical strain measurement points are labeled No. 1, No. 3 and No. 5 and the transverse strain measurement points of the corresponding specimens are labeled as No. 2, No. 4 and No. 6. The load–strain curves are shown in Figure 12. It is obtained from the Figure 12 that the strain value increases with the increase of axial compressive load. When the axial compressive strength increased within the elastic stage, the strain is linearly increased with the increase of axial compressive strength. When the axial compressive strength increases beyond the elastic stage, the increased rate of strain is greater than that within the elastic stage.

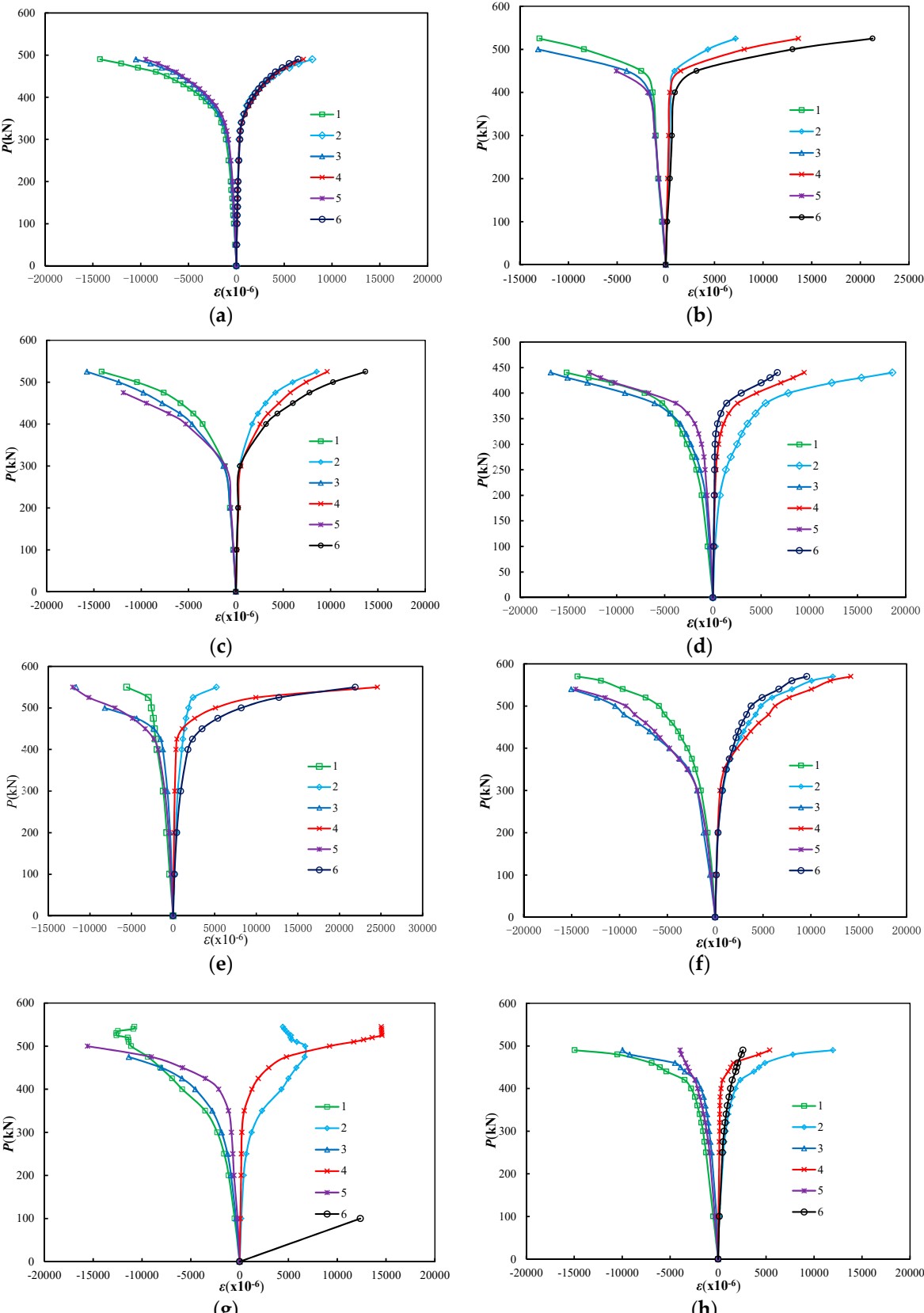

**Figure 12.** *Cont.*

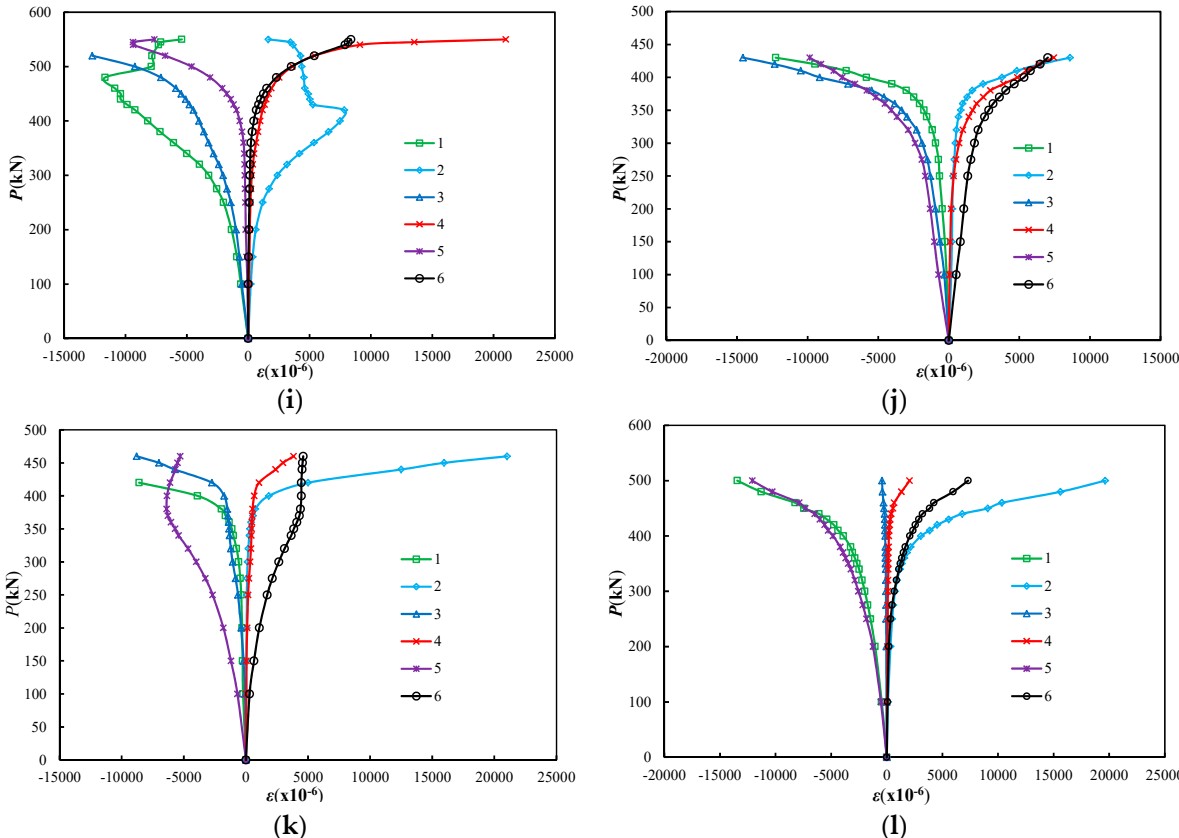

**Figure 12.** Load–strain curves: (**a**) C20-L0-E0; (**b**) C30-L0-E0; (**c**) C40-L0-E0; (**d**) C20-L0.50-E10000; (**e**) C30-L0.50-E5000; (**f**) C30-L0.50-E7500; (**g**) C30-L0.50-E10000; (**h**) C30-L0.50-E12500; (**i**) C30-L0.50-E15000; (**j**) C30-L0.25-E10000; (**k**) C30-L0.33-E10000; (**l**) C40-L0.50-E10000.

By comparing the load-strain curves of different specimens without the damage of lateral impact, the increased rates of strain are mostly identical when axial compressive load increases within the elastic stage, elastic–plastic stage and strengthened stage. For the most specimens with damage of lateral impact, the increased rate of strain is greater at impact location than those at other strain measurement points within the elastic stage. Increased rates of strain at the No. 3 and No. 4 measurement points are greater than those at the No. 5 and No. 6 measurement points. When axial compressive load increases within the elastic–plastic stage and strengthened stage, the increased rate of strain at the impact location is greater than that the within elastic stage. The increased rate of strain at the impact location is greater than the increased rate of strain at the other measurement points. Results indicate that the steel at the impact location becomes plastic faster than other locations and load–strain curves of measurement points 1 and 2 are symmetrical. It also indicates that increased rate of strain at vertical strain measurement point and transverse strain measurement point are mostly the same. Because of the complicated deformation, load–strain curves show a turn back tendency for specimens C30-L0.50-E10000, C30-L0.50-E15000 and C30-L0.33-E10000 when the axial compressive strength increases within the later load stage. The curves of vertical strain and transverse strain of these specimens do not have a clear trend. Results indicate that compared with other locations, impact location damage becomes serious when the columns are subjected to the lateral impact. So the reinforcement of possible impact location should be taken into account when the CFST columns are designed in the civil engineering project.

### 3.4. Effects of Impact Energy

Curves of load-displacement are shown in Figure 13a with different impact energy levels. The corresponding initial stiffness and ductility are shown in Table 5. At the early stage of increasing axial compressive load, the load-displacement curves for specimens subjected to greater impact energy are more placid than those of other specimens. When the compressive strength of concrete is 20 MPa and the impact location is at 0.25 *H*, impact energies are 0 *J*, 5000 *J*, 7500 *J*, 10,000 *J*, 12,500 *J* and 15,000 *J* and the corresponding initial stiffnesses are 1.00, 0.68, 0.67, 0.65, 0.47 and 0.90 respectively. Lateral impact has negative effects on the behavior of circular CFST columns indicating that initial stiffness decreases as impact energy increases. When the axial compressive load increases within the later load stage, the load–displacement curves are placid. The ductilities of the aforementioned specimens are 3.53, 2.05, 2.21, 2.78, 2.34 and 2.00. It shows that ductility of specimen decreases when the impact energy increases, and the residual ultimate axial capacity decreases when the impact energy increases.

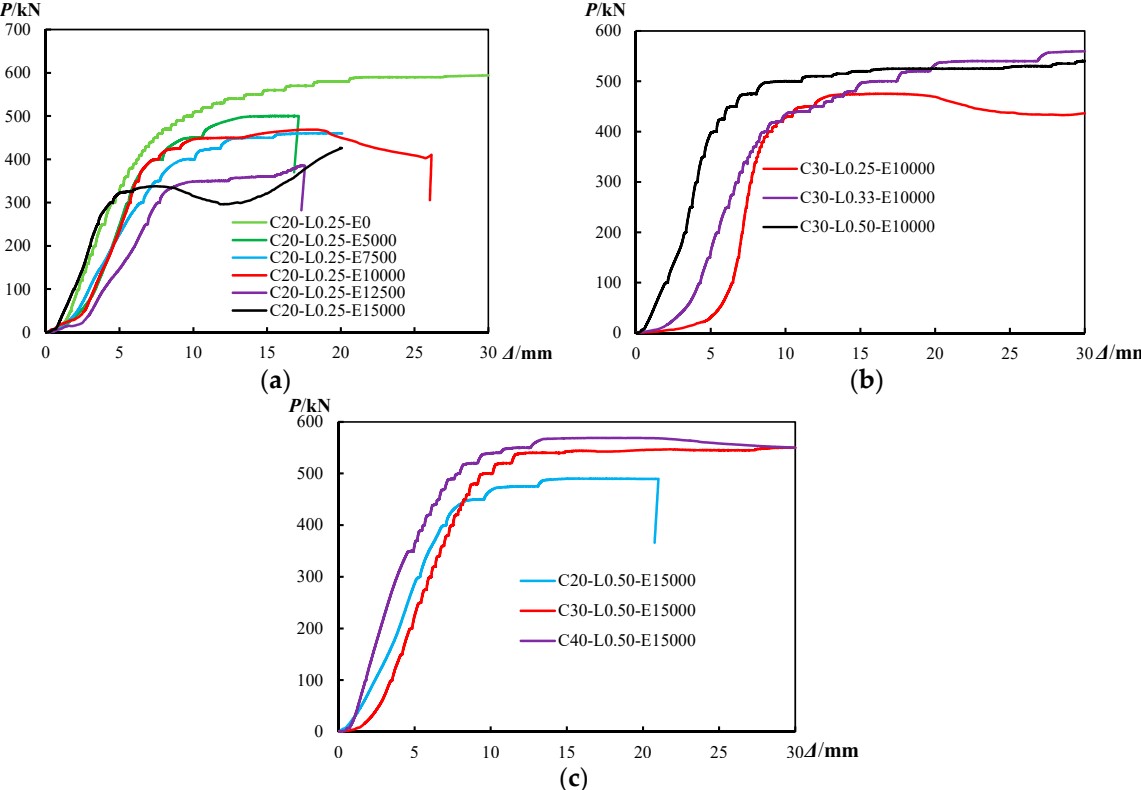

**Figure 13.** Comparison of load-displacement curves: (**a**) different impact energy; (**b**) different impact location; (**c**) different concrete compressive strength.

The *ρ*-impact energy curves are presented in Figure 14 with different impact locations. When the impact location and the compressive strength of concrete are constant, the value of *ρ* decreases as the impact energy increases, which indicates that residual ultimate axial capacity decreases when impact energy increases. When the concrete compressive strength is 20 MPa and the impact location is at 0.25 *H*, the residual ultimate axial capacity decreases by 43%, which can be obtained from Figure 14c.

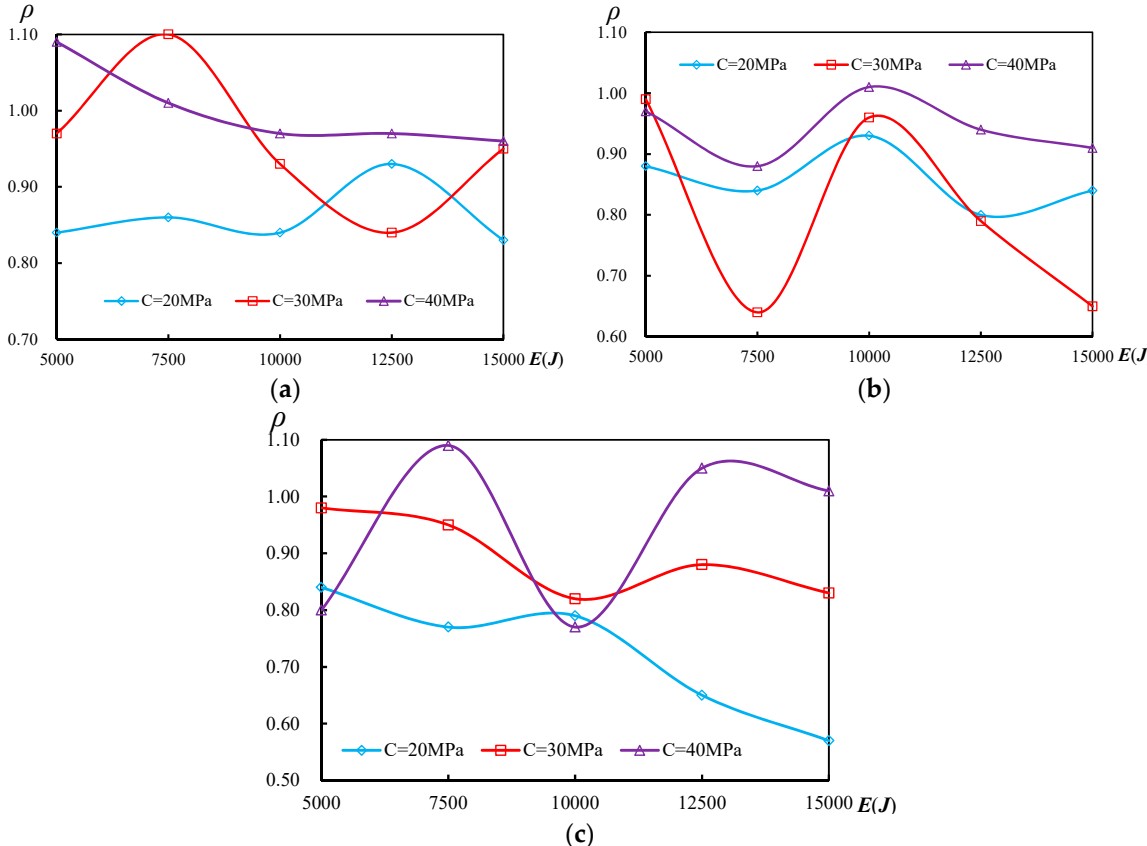

**Figure 14.** Curves of $\rho$-impact energy: (**a**) $L = 0.50\ H$; (**b**) $L = 0.33\ H$; (**c**) $L = 0.25\ H$.

### 3.5. Effects of Impact Location

Figure 13b demonstrates the difference of load–displacement curves at different impact locations. When the axial compressive load increases within early stage, load–displacement curves of specimen, whose impact location approaches the end of specimen, are more placid than that of other specimens. When the impact energy and the compressive strength of concrete are constant, initial stiffness decreases with impact location approaching the specimen end. For example, when concrete compressive strength is 30 MPa and impact energy is 10,000 J, impact location is moved from 0.50 H to 0.25 H and corresponding initial stiffness decreases from 1.13 to 0.45. This indicates that the closer to the specimen end the impact location is, the greater the initial stiffness decreases. The ductility of these above specimens decreases from 5.05 to 1.85 when the impact location is moved from 0.50 H to 0.25 H at the later stage of increasing axial compressive load. In addition, the ductility decreases as the impact location gets close to the end of the specimen. The residual ultimate axial capacity decreases with the impact location approaching the specimen end, which is summarized in Table 5 and Figure 11.

Curves of $\rho$-impact location are presented in Figure 15 with different compressive strength of concrete which indicates that the value of $\rho$ decreases when impact location comes close to the specimen end. From Figure 15a the value of $\rho$ decreases by 35% (C20-E12500), indicating that residual ultimate axial capacity decreased as impact location gets close to the specimen end. This trend is more obvious in Figure 15a than that of Figure 15c, which shows that improving concrete compressive strength reduces the negative effects of impact location on residual ultimate axial capacity. The damage is more serious when at the end of specimens subjected to lateral impact than at other locations. When CFST columns are designed, the end of the column should be strengthened to improve the columns' impact resistance.

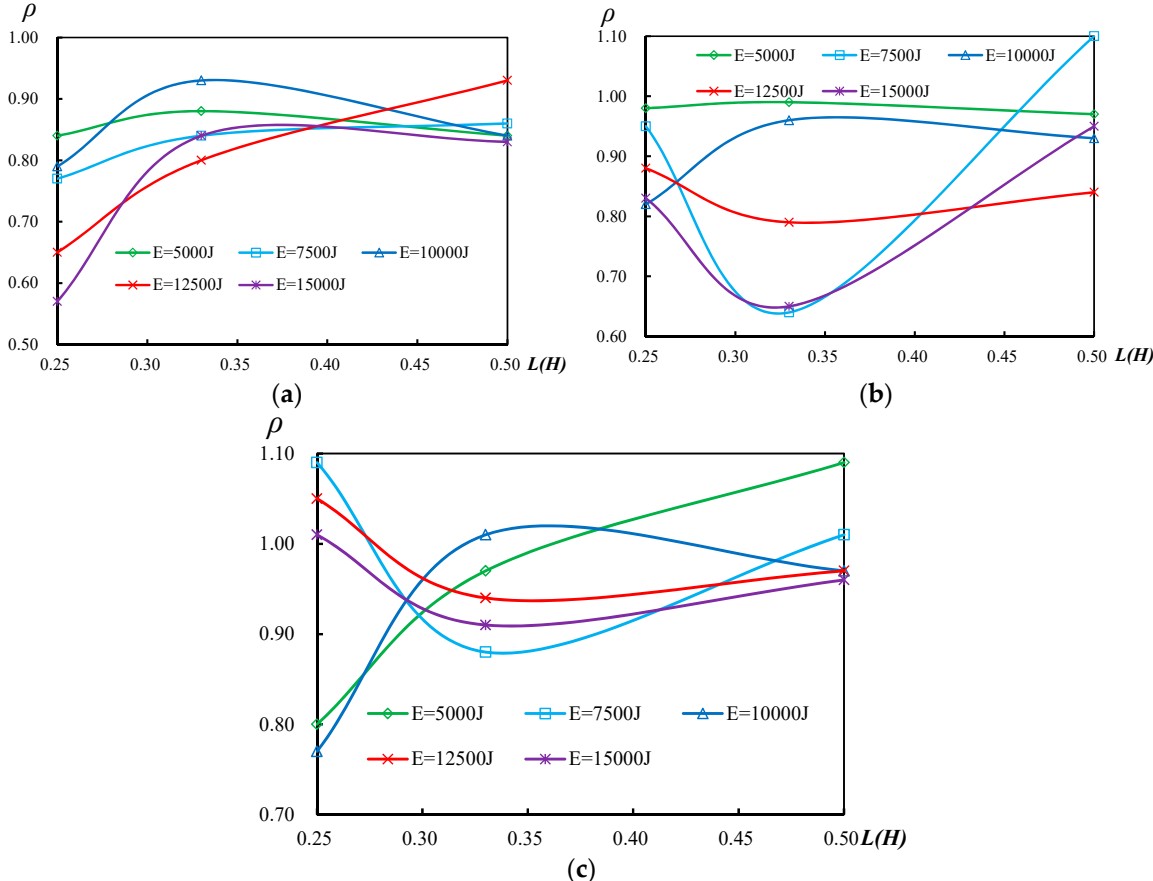

**Figure 15.** Curves of ρ-impact location: (**a**) concrete compressive strength is 20 MPa; (**b**) concrete compressive strength is 30 MPa; (**c**) concrete compressive strength is 40 MPa.

### 3.6. Effects of Concrete Compressive Strength

Concrete compressive strength effects on the residual ultimate axial capacity are presented in Figure 13c, which shows the curves of load–displacement with different compressive strength of concrete. At the early stage of increasing axial compressive load, the load-displacement curves of specimens with lower compressive strength of concrete are more placid than those of other specimens. Initial stiffness of these specimens C40-L0.50-E15000, C30-L0.50-E15000 and C20-L0.50-E15000 are 1.76, 0.79 and 0.78, respectively. This indicates that initial stiffness increases with the increasement of concrete compressive strength. In other words, the lower the concrete compressive strength, the more serious the loss of initial stiffness. The curves of all specimens are placid when axial compressive load is increased within the strengthened stage. Ductility of specimens subjected to lateral impact damage has an increasing trend as the concrete compressive strength increases. However, when compressive strength of concrete is 30 MPa, the ductility is the greatest among all specimens with different compressive strength of concrete. Based on the discussion above, it is suggested that the concrete compressive strength of 30 MPa be used in civil engineering projects. From Table 5 and Figure 11, it can be seen that ductility and ultimate axial capacity decrease when the compressive strength of concrete decreases.

Curves of ρ-concrete compressive strength with different impact locations are shown in Figure 16, which shows that the value of ρ decreases when concrete compressive strength decreases. In other words, ultimate axial capacity decreases as the compressive strength of concrete decreases. The trend is more obvious in Figure 16c than that in Figure 16a,b, illustrating that the damage of lateral impact is more serious at 0.25 *H* than that of other impact location.

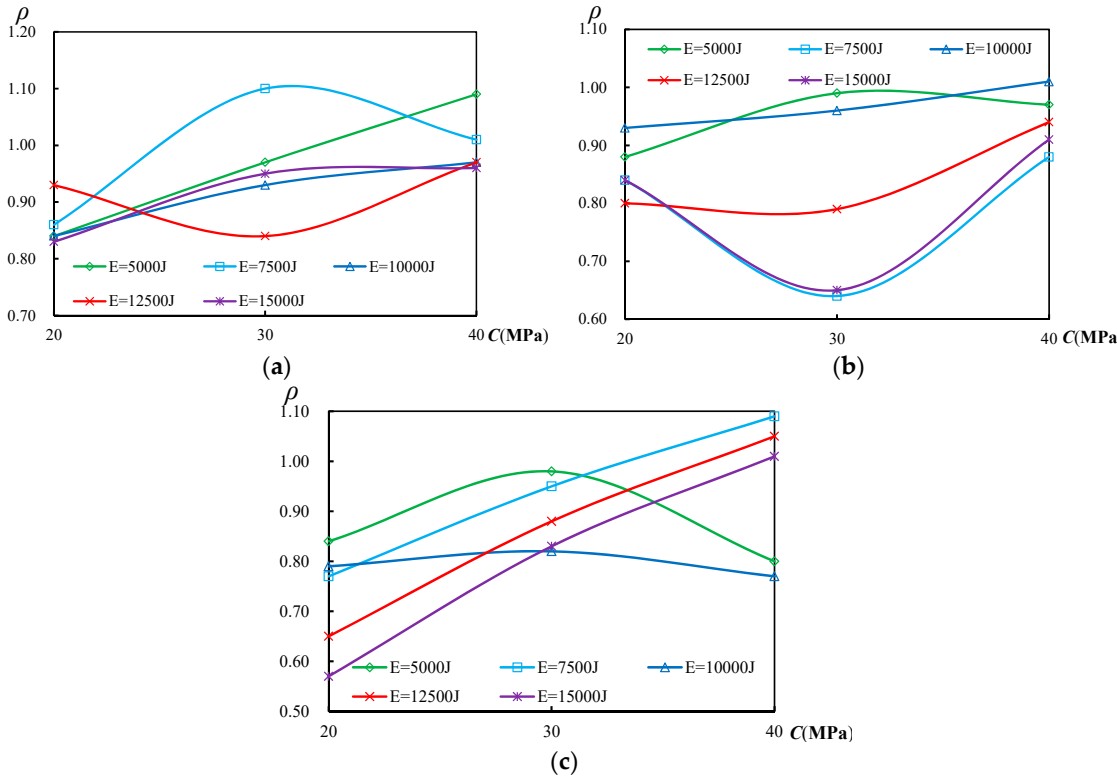

**Figure 16.** Curves of $\rho$-concrete compressive strength: (**a**) Impact location is 0.50 H; (**b**) Impact location is 0.33 H; (**c**) Impact location is 0.25 H.

## 4. Calculation Formula

This paper has investigated the effects of these three experimental parameters (concrete compressive strength, impact energy and impact location) on the behavior of circular CFST columns subjected to lateral impact. The results find that the comprehensive corrected coefficient ($\varphi$) determined by the experimental parameters influences residual ultimate axial capacity. Hence, residual ultimate axial capacity ($N_{re}$) can be obtained by the following Equation (4):

$$N_{re} = \varphi \times N_0 \tag{3}$$

where, $N_{re}$ is calculated residual ultimate axial capacity of circular CFST columns subjected to lateral impact, and $N_0$ is calculated ultimate axial capacity of circular CFST columns immune from the damage of lateral impact. The value can be obtained from the following Equations [39,40]:

$$N_0 = A_{sc} \times f_{scy} \tag{4}$$

$$f_{scy} = (1.14 + 1.02\xi) \times f_{cu} \tag{5}$$

$$A_{sc} = A_c + A_s \tag{6}$$

$$\xi = \alpha \times f_y / f_{cu} \ (0.8 \leq \xi \leq 4) \tag{7}$$

$$\alpha = A_s / A_c \tag{8}$$

where, $A_s$ is area of the steel tube in cross-section, $A_c$ is area of concrete in cross-section, $f_y$ is the yield strength of steel tube, and $f_{cu}$ is the compressive strength of concrete.

According to Equation (3), the comprehensive corrected coefficient ($\varphi$) could be calculated by introducing it into the effected parameter $M$ as following:

$$\varphi = 1.23M + 0.026 \tag{9}$$

where, effected parameter $M$ has taken into account the effects of concrete compressive strength ($f(\xi)$), impact location ($f(\alpha)$) and impact energy ($f(\beta)$). The value of $M$ is calculated as follows:

$$M = f(\xi) \times f(\alpha) \times f(\beta) \tag{10}$$

$$f(\xi) = -0.123 \times \xi + 1.08 \tag{11}$$

$$f(\alpha) = 0.36 \times \alpha + 0.76 \tag{12}$$

$$f(\beta) = -0.043 \times \beta + 0.972 \tag{13}$$

$$\alpha = L/H \tag{14}$$

$$\beta = E/5000 \tag{15}$$

where, $f(\xi)$, $f(\alpha)$ and $f(\beta)$ are function of $\xi$, $\delta$ and $\beta$, respectively. Impact location is represented by $L$ and impact energy is represented by $E$.

The calculated and statistical results are presented in Table 6 and Figure 17. Table 6 presents the values ($\lambda$) of ratio of $N_{re}$ (residual ultimate axial capacity calculated according to equation) to $N_r$ (residual ultimate axial capacity obtained by test). The most calculated residual ultimate axial capacity values of circular CFST columns subjected to lateral impact are slightly smaller than residual ultimate axial capacity value of test results. This indicates that the calculated result satisfying the requirement of practical engineering has a strength reservation. The mean value ($\lambda$) is 0.97 and the variance value is 0.01365, which indicates that this equation is reasonably accurate to predict residual ultimate axial capacity of circular CFST columns subjected to lateral impact.

**Table 6.** Comparison of residual ultimate axial capacity between formula and experiment.

| Specimen | $f_y$ (MPa) | $f_u$ (MPa) | $f_{cu}$ (MPa) | $\xi$ | $\alpha$ | $\beta$ | $N_r$ (kN) | $N_{re}$ (kN) | $\rho$ ($N_{re}/N_r$) | Error (%) |
|---|---|---|---|---|---|---|---|---|---|---|
| C20-L0-E0 | 264 | 352 | 22.13 | 2.51 | – | – | 594.10 | – | – | – |
| C20-L0.50-E5000 | 264 | 352 | 22.13 | 2.51 | 0.50 | 1.00 | 500.15 | 508.74 | 1.02 | 2 |
| C20-L0.50-E7500 | 264 | 352 | 22.13 | 2.51 | 0.50 | 1.50 | 510.00 | 495.33 | 0.97 | −3 |
| C20-L0.50-E10000 | 264 | 352 | 22.13 | 2.51 | 0.50 | 2.00 | 500.05 | 485.81 | 0.97 | −3 |
| C20-L0.50-E12500 | 264 | 352 | 22.13 | 2.51 | 0.50 | 2.50 | 550.20 | 471.46 | 0.86 | −14 |
| C20-L0.50-E15000 | 264 | 352 | 22.13 | 2.51 | 0.50 | 3.00 | 490.20 | 459.33 | 0.94 | −6 |
| C20-L0.33-E5000 | 264 | 352 | 22.13 | 2.51 | 0.33 | 1.00 | 523.75 | 476.43 | 0.91 | −9 |
| C20-L0.33-E7500 | 264 | 352 | 22.13 | 2.51 | 0.33 | 1.50 | 500.30 | 466.09 | 0.93 | −7 |
| C20-L0.33-E10000 | 264 | 352 | 22.13 | 2.51 | 0.33 | 2.00 | 550.10 | 452.29 | 0.82 | −18 |
| C20-L0.33-E12500 | 264 | 352 | 22.13 | 2.51 | 0.33 | 2.50 | 475.25 | 443.60 | 0.93 | −7 |
| C20-L0.33-E15000 | 264 | 352 | 22.13 | 2.51 | 0.33 | 3.00 | 500.25 | 434.02 | 0.87 | −13 |
| C20-L0.25-E5000 | 264 | 352 | 22.13 | 2.51 | 0.25 | 1.00 | 500.30 | 461.65 | 0.92 | −8 |
| C20-L0.25-E7500 | 264 | 352 | 22.13 | 2.51 | 0.25 | 1.50 | 460.25 | 452.94 | 0.98 | −2 |
| C20-L0.25-E10000 | 264 | 352 | 22.13 | 2.51 | 0.25 | 2.00 | 468.70 | 439.29 | 0.94 | −6 |
| C20-L0.25-E12500 | 264 | 352 | 22.13 | 2.51 | 0.25 | 2.50 | 385.20 | 428.52 | 1.11 | 11 |
| C20-L0.25-E15000 | 264 | 352 | 22.13 | 2.51 | 0.25 | 3.00 | 337.85 | 418.32 | 1.24 | 24 |
| C30-L0-E0 | 264 | 352 | 32.70 | 1.70 | – | – | 580.00 | – | – | – |
| C30-L0.50-E5000 | 264 | 352 | 32.70 | 1.70 | 0.50 | 1.00 | 563.10 | 558.13 | 0.99 | −1 |
| C30-L0.50-E7500 | 264 | 352 | 32.70 | 1.70 | 0.50 | 1.50 | 640.00 | 546.79 | 0.85 | −15 |
| C30-L0.50-E10000 | 264 | 352 | 32.70 | 1.70 | 0.50 | 2.00 | 540.10 | 533.21 | 0.99 | −1 |
| C30-L0.50-E12500 | 264 | 352 | 32.70 | 1.70 | 0.50 | 2.50 | 486.80 | 519.54 | 1.07 | 7 |
| C30-L0.50-E15000 | 264 | 352 | 32.70 | 1.70 | 0.50 | 3.00 | 550.00 | 506.49 | 0.92 | −8 |
| C30-L0.33-E5000 | 264 | 352 | 32.70 | 1.70 | 0.33 | 1.00 | 575.10 | 523.13 | 0.91 | −9 |
| C30-L0.33-E7500 | 264 | 352 | 32.70 | 1.70 | 0.33 | 1.50 | 370.05 | 508.99 | 1.38 | 38 |
| C30-L0.33-E10000 | 264 | 352 | 32.70 | 1.70 | 0.33 | 2.00 | 559.50 | 501.25 | 0.90 | −10 |
| C30-L0.33-E12500 | 264 | 352 | 32.70 | 1.70 | 0.33 | 2.50 | 460.00 | 489.01 | 1.06 | 6 |
| C30-L0.33-E15000 | 264 | 352 | 32.70 | 1.70 | 0.33 | 3.00 | 379.45 | 478.45 | 1.26 | 26 |
| C30-L0.25-E5000 | 264 | 352 | 32.70 | 1.70 | 0.25 | 1.00 | 566.75 | 504.22 | 0.89 | −11 |
| C30-L0.25-E7500 | 264 | 352 | 32.70 | 1.70 | 0.25 | 1.50 | 550.00 | 493.44 | 0.90 | −10 |
| C30-L0.25-E10000 | 264 | 352 | 32.70 | 1.70 | 0.25 | 2.00 | 474.95 | 482.32 | 1.02 | 2 |

**Table 6.** *Cont.*

| Specimen | $f_y$ (MPa) | $f_u$ (MPa) | $f_{cu}$ (MPa) | $\xi$ | $\alpha$ | $\beta$ | $N_r$ (kN) | $N_{re}$ (kN) | $\rho$ ($N_{re}/N_r$) | Error (%) |
|---|---|---|---|---|---|---|---|---|---|---|
| C30-L0.25-E12500 | 264 | 352 | 32.70 | 1.70 | 0.25 | 2.50 | 510.25 | 471.49 | 0.92 | −8 |
| C30-L0.25-E15000 | 264 | 352 | 32.70 | 1.70 | 0.25 | 3.00 | 479.70 | 458.65 | 0.96 | −4 |
| C40-L0-E0 | 264 | 352 | 43.13 | 1.29 | – | – | 595.00 | – | – | – |
| C40-L0.50-E5000 | 264 | 352 | 43.13 | 1.29 | 0.50 | 1.00 | 648.80 | 604.52 | 0.93 | −7 |
| C40-L0.50-E7500 | 264 | 352 | 43.13 | 1.29 | 0.50 | 1.50 | 598.95 | 588.69 | 0.98 | −2 |
| C40-L0.50-E10000 | 264 | 352 | 43.13 | 1.29 | 0.50 | 2.00 | 579.50 | 579.38 | 0.99 | −1 |
| C40-L0.50-E12500 | 264 | 352 | 43.13 | 1.29 | 0.50 | 2.50 | 576.75 | 563.02 | 0.98 | −2 |
| C40-L0.50-E15000 | 264 | 352 | 43.13 | 1.29 | 0.50 | 3.00 | 569.00 | 547.66 | 0.96 | −4 |
| C40-L0.33-E5000 | 264 | 352 | 43.13 | 1.29 | 0.33 | 1.00 | 575.25 | 564.09 | 0.98 | −2 |
| C40-L0.33-E7500 | 264 | 352 | 43.13 | 1.29 | 0.33 | 1.50 | 526.25 | 556.01 | 1.06 | 6 |
| C40-L0.33-E10000 | 264 | 352 | 43.13 | 1.29 | 0.33 | 2.00 | 600.85 | 540.38 | 0.90 | −10 |
| C40-L0.33-E12500 | 264 | 352 | 43.13 | 1.29 | 0.33 | 2.50 | 560.00 | 528.39 | 0.94 | −6 |
| C40-L0.33-E15000 | 264 | 352 | 43.13 | 1.29 | 0.33 | 3.00 | 540.10 | 513.71 | 0.95 | −5 |
| C40-L0.25-E5000 | 264 | 352 | 43.13 | 1.29 | 0.25 | 1.00 | 475.75 | 547.62 | 1.15 | 15 |
| C40-L0.25-E7500 | 264 | 352 | 43.13 | 1.29 | 0.25 | 1.50 | 650.00 | 536.79 | 0.83 | −17 |
| C40-L0.25-E10000 | 264 | 352 | 43.13 | 1.29 | 0.25 | 2.00 | 460.25 | 525.67 | 1.14 | 14 |
| C40-L0.25-E12500 | 264 | 352 | 43.13 | 1.29 | 0.25 | 2.50 | 622.15 | 508.82 | 0.82 | −18 |
| C40-L0.25-E15000 | 264 | 352 | 43.13 | 1.29 | 0.25 | 3.00 | 600.00 | 497.83 | 0.83 | −17 |
| Mean value | | | | | | | | | 0.97 | −3 |
| Minimum value | | | | | | | | | 0.82 | −18 |
| Maximum value | | | | | | | | | 1.38 | −38 |
| Variance | | | | | | | | | 0.0136 | 0.0136 |

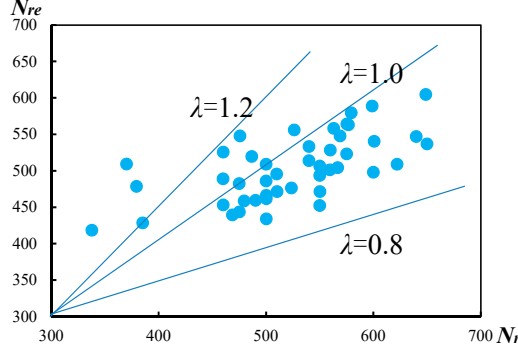

**Figure 17.** Statistic result of $\lambda$.

## 5. Conclusions

In this paper, the effects of compressive strength of concrete and lateral impact on the residual ultimate axial capacity of circular CFST columns subjected to lateral impact have been studied. Residual ultimate axial capacity, failure modes, ductility and initial stiffness of circular CFST columns subjected to lateral impact have been analyzed. The following conclusions can be obtained from this test:

(1) By analyzing the axial compression test process, four kinds of failure modes are summarized in this paper: the first bulged position occurs at the specimen end, which is opposed to the impact point; the first bulged position occurs at the impact point; the bulged position occurs at the impact point, and then deformation is enlarged when the axial compressive load increased. The bulged position occurs at the specimen end, which is opposed to the impact point, and the bulged position develops a ringed bulge as the axial compressive load increased.

(2) Lateral impact has negative effects on residual ultimate axial capacity. Residual ultimate axial capacity decreases as impact location gets close to the end of specimen. In addition, improving concrete compressive strength can reduce the negative effects of impact location on residual ultimate axial capacity.

(3) Ductility and initial stiffness of the specimens are affected by lateral impact. Initial stiffness and ductility decrease when impact energy increases. Initial stiffness and ductility decrease when impact location gets close to the specimen end. When the compressive strength of concrete is 30 MPa, ductility is greater than that of other specimens with different compressive strengths of concrete.

(4) The value of $N_{\mathrm{re}}$ obtained from the equation is reasonable to predict residual ultimate axial capacity of circular CFST columns, which are subjected to the damage of lateral impact. This equation obtaining the value of $N_{\mathrm{re}}$ is verified to be accurate.

**Author Contributions:** All authors discussed and agreed upon the idea, and made scientific contributions. X.Z., and Y.C. designed the experiments and wrote the paper. X.Z., X.S., and Y.Z. performed the experiments and analyzed the data.

**Acknowledgments:** This research was founded by the Excellent Doctoral and Master's Thesis Cultivation Founded Program of Yangtze University, the National Natural Science Foundation of China (No. 51778066) and Hubei Province Outstanding Youth Science Foundation of China (No. 2017CFA070).

**Conflicts of Interest:** The authors declare no conflict of interest.

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
