# Peer review of "Behavior of Circular CFST Columns Subjected to Different Lateral Impact Energy"

_applsci, doi:10.3390/app9061134_

Round 1

Reviewer 1 Report

The topic of concrete filled steel tube columns is important. The authors are to be commended
fore taking such an extensive effort.

The authors are advised to talk to an expert English writer. This may help improve
the language part of the paper.

Author Response

Response to Reviewer 1 Comments

Point 1: The topic of concrete filled steel tube columns is important. The authors are to be commended fore taking such an extensive effort. The authors are advised to talk to an expert English writer. This may help improve the language part of the paper.

Response 1: Many thanks to the reviewer 1 for the positive comments. The authors have sought the advised from experienced writers and the poor part has been revised in the manuscript.

Reviewer 2 Report

I think that the paper and the experimental resulsts are interesting. However, I believe that the paper can be improved if the Authors provide a more mature discussion of the results with a direct link to practical design against impact.

Author Response

Response to Reviewer 2 Comments

Point 1: I think that the paper and the experimental resulsts are interesting. However, I believe that the paper can be improved if the Authors provide a more mature discussion of the results with a direct link to practical design against impact.

Response 1: The authors agree with the reviewer. The authors have added some discussion on enhancing the connection of test results with the practical design against impact in the manuscript. The authors have given advices about improving the impact resistance of CFST columns from the test results (failure modes and effects of investigated parameters).

Reviewer 3 Report

the topic and the paper are interesting, especially the rexperimental results. I have only a minor remark regarding the interpretation and the use of the experimental results. Indeed. although the formula, some additional discussion on the implications for the design of such structural elements under impact should be done. For instance, a design criterion with a flowchart may be useful.

Author Response

Response to Reviewer 3 Comments

Point 1: the topic and the paper are interesting, especially the rexperimental results. I have only a minor remark regarding the interpretation and the use of the experimental results. Indeed. although the formula, some additional discussion on the implications for the design of such structural elements under impact should be done. For instance, a design criterion with a flowchart may be useful.

Response 1: The authors thanks for the patient review and pertinent comments of the reviewer 3. The authors have added some discussion on the implications for the design of CFST columns under impact from the reason of test parameters choice, connection of test result with practical design against impact and so on.

This manuscript is a resubmission of an earlier submission. The following is a list of the peer review reports and author responses from that submission.